# Broadband Circularly Polarized Conical Corrugated Horn Antenna Using a Dielectric Circular Polarizer

**DOI:** 10.3390/mi13122138

**Published:** 2022-12-03

**Authors:** Jun Xiao, Jin Tian, Tongyu Ding, Hongmei Li, Qiubo Ye

**Affiliations:** 1School of Ocean Information Engineering, Jimei University, Xiamen 361021, China; 2School of Electronics and Information Engineering, Harbin Institute of Technology, Harbin 150001, China

**Keywords:** circular polarization (CP), corrugated horn antenna, circular polarizer, broadband

## Abstract

In this paper, a broadband left-handed circularly polarized (LHCP) corrugated horn antenna using a dielectric circular polarizer is proposed. Circularly polarized (CP) waves are generated by inserting an improved dovetail-shaped dielectric plate into the circular waveguide. Compared with the traditional dovetail-shaped circular polarizer, the proposed improved dovetail-shaped circular polarizer has a wider impedance bandwidth and 3 dB axial ratio bandwidth. A substrate-integrated waveguide (SIW) structure is designed as a wall to eliminate the influence of fixed grooves on the circular polarizer. The simulated reflection coefficient of the dielectric plate circular polarizer is less than −20 dB in the frequency band from 17.57 to 33.25 GHz. Then, a conical corrugated horn antenna with five corrugations and a four-level metal stepped rectangular-circular waveguide converter are designed and optimized. The simulated −10 dB impedance and 3 dB axial ratio (AR) bandwidths of the circularly polarized horn antenna integrated with the polarizer are 61% (17.1–32.8 GHz) and 60.9% (17.76–33.32 GHz), respectively. The simulated peak gain is 17.34 dBic. The measured −10 dB impedance is 52.7% (17.2–27.5 GHz).

## 1. Introduction

Circularly polarized (CP) horn antennas are widely used in many fields, such as satellite communications, radar and radio astronomy [1], due to their wide bandwidth, symmetrical radiation pattern and mitigation of polarization mismatch [2]. The operating band of CP horn antennas has been extended to millimeter wave (MMW) [3,4,5] and even terahertz (THz) bands [6,7,8,9]. Many circularly polarized horn antennas operating in dual bands have also been proposed [10,11,12,13]. 

A representative CP horn antenna essentially consists of a circular polarizer and a horn antenna [14]. Corrugated horn antennas are one of the most popular feed antennas due to their advantages, such as low cross-polarization, symmetrical radiation patterns in two orthogonal planes and low sidelobe level [15,16,17,18,19,20,21]. A circular polarizer is an important part of the circularly polarized horn antenna and can convert linearly polarized waves into circularly polarized waves.

In previous research works, various waveguide polarizers have been proposed for feed horn antennas, and these polarizers can be divided into three types in general. The first type is to create grooves [22] or irises [23] in the waveguide walls as a way to achieve orthogonal polarization waves. This type of polarizer attains a high strength structure, a wide bandwidth and a low loss. Its disadvantages are clear, as it is extremely difficult to manufacture the grooves and irises for MMW and THz fabrications. The second type is the septum polarizer [24,25,26,27,28], which is more compact compared to the other two polarizers. Its typical structure comprises a metal septum at the center of a square waveguide. The septum polarizer has a simple structure; however, because it is asymmetric, it can impair the performance of the horn antenna [29]. 

The last type of circular polarizer is achieved by filling a dielectric plate inside the waveguide [30,31]. This circular polarizer has a wide operating bandwidth and does not affect the performance of the horn antenna because it is a symmetrical structure. Compared with other waveguide circular polarizers, dielectric plate circular polarizers are simpler to process, easier to debug and more widely used.

In this paper, a circularly polarized conical corrugated horn antenna loaded with a dielectric plate circular polarizer is presented. The proposed CP horn antenna consists of three parts: a dielectric plate circular polarizer, a conical corrugated horn antenna and a four-level metal stepped waveguide converter. Among them, the dielectric plate circular polarizer is the key to generating circularly polarized waves. This design of a circular polarizer improves the performance of circular polarization by changing the shape of the dielectric plate. 

Previous designs eliminated the influence of the fixed grooves on the circular polarizer by digging compensation grooves in the plane orthogonal to the fixed grooves. This design proposes a substrate-integrated waveguide (SIW) wall to eliminate the influence of the fixed grooves on the circular polarizer. A conical corrugated horn antenna with five corrugations was designed as the radiating part. 

The simulated 10 dB impedance and 3 dB AR bandwidths of the proposed circularly polarized conical corrugated horn antenna were 61% (17.1–32.8 GHz) and 60.9% (17.76–33.32 GHz), respectively. The simulated peak gain was 17.34 dBic. The measured 10 dB impedance was 52.7% (17.2–27.5 GHz). The proposed circularly polarized conical corrugated horn antenna system is a promising candidate for future 5G applications. The specifications of sub −6 GHz 5G applications are explained in [32,33].

## 2. Antenna Design

### 2.1. Configuration of the CP Horn Antenna

Figure 1 shows the overall structure of the proposed circular polarized horn antenna. The proposed CP horn antenna consists of three parts: a circular polarizer by filling an improved dovetail-shaped dielectric plate in the waveguide, a conical corrugated horn antenna and a four-level metal stepped rectangular-circular waveguide converter.

### 2.2. Dielectric Plate Circular Polarizer Design

The waveguide circular polarizer is an essential part of the circularly polarized horn antenna and directly affects the performance of the circularly polarized horn antenna. For dielectric plate circular polarizers, the dielectric plate is used as a phase-shifting element. The dielectric plate generates two effective dielectric constants for two orthogonal polarization waves parallel and perpendicular to it, respectively. The phase difference can be obtained by changing the structure of the dielectric plate. In this way, two orthogonally polarized waves can be obtained. The dielectric plate circular polarizer has the advantages of simple structure, convenient processing and good adjustability.

Figure 2 shows the proposed dielectric plate circular polarizer. The dielectric material used in this design is Rogers RT5880 with a relative dielectric constant of 2.2. This circular polarizer consists of three parts: a circular waveguide, an improved dovetail dielectric plate and a SIW wall. The dielectric plate is placed in the center of the circular waveguide at an angle of 45° to the incident wave *E* of the linear polarization. When *E* is incident from port 1, it can be decomposed into two equal-amplitude, in-phase orthogonal polarization components *E_x_* and *E_y_*, where *E_x_* is parallel to the dielectric plate and *E_y_* is perpendicular to the dielectric plate. 

When *E_x_* and *E_y_* pass through the dielectric plate, the dielectric plate produces different phase shift constants for the two orthogonal polarization components, and a 90° phase difference of the *E_x_* and *E_y_* waves can be produced by adjusting the structure of the dielectric plate. The dielectric plate is fixed inside the circular waveguide by digging fixing grooves in the inner wall of the circular waveguide. It can be seen from reference [30] that the performance of the circular polarizer is affected by the fixed grooves on the circular waveguide wall. In reference [31], it was proposed to dig compensation grooves in a plane orthogonal to the fixed grooves to eliminate the influence of the fixed grooves on the circular polarizer. 

As the depth of the compensation grooves has a clear influence on the circular polarizer, high machining accuracy is required, particularly in the millimeter band. In the proposed circular polarizer, a SIW structure was designed as a wall to eliminate the influence of fixed grooves on the circular polarizer. Moreover, the high design flexibility of SIW is beneficial to the high-performance circular polarizers. The incident wave is incident from port 1 and is emitted from port 2 through the dielectric plate forming a circularly polarized wave.

This design proposed an improved dovetail-shaped dielectric plate as shown in Figure 3c based on the common rectangular shape and traditional dovetail-shaped dielectric plate. The conventional dovetail structure is formed by two straight lines and a pointed tip. This proposed design in Figure 3c is formed by two smooth curves, and the pointed tip in Figure 3b is changed to flat. 

Taking the derivative of both sides of (1) shows that the derivative has zeros as *x* = 0 and *x* = *l*. Compared with the traditional dovetail-shaped dielectric circular polarizer, the designed improved dovetail-shaped dielectric circular polarizer can reduce the generation of high-order mode, and the performance of the improved dovetail-shaped circular polarizer is better. Figure 4 shows the simulated |S_11_| for three different shapes of dielectric plates. From the simulated results, it can be seen that the improved dovetail-shaped structure has better |S_11_|.

For the improved dovetail-shaped structure in Figure 3c, one side curve is given by
(1)$$\begin{array}{l} y=(a_{2}-a_{1})\cdot \sin^{2}(\frac{\pi x}{2l})+a_{1}\\ y^{\prime }=\frac{\pi (a_{2}-a_{1})}{2l}\cdot \sin(\frac{\pi x}{l}) \end{array}$$
where *a*_2_ is the diameter of the inner wall of the circular waveguide, *a*_1_ is half of the length of flat (*w*_4_/2) and x∈[0,l].

Figure 5 shows the simulated results of the dielectric plate circular polarizer. It can be seen in Figure 5a that the simulated bandwidth of the dielectric plate circular polarizer is from 17.5 to 32.5 GHz for |S_11_| ≤ −20 dB. As can be seen from Figure 5b that the simulated AR bandwidth of the dielectric plate circular polarizer is from 17.4 to 32.5 GHz for AR ≤ 3 dB. As shown in Figure 5c, the simulated results of *E*-field distributions indicate good circular polarization performance. 

According to Figure 5, it can be predicted that the horn antenna loaded with the improved dovetail-shaped circular polarizer will have good circular polarization performance. To explain the mechanism of the circular polarization, the current distributions on port 2 at phases of 0°, 90°, 180° and 270° are illustrated in Figure 6. It can be seen that, as the time varies, the electrical currents on the port 2 rotate in the clockwise direction, and hence a left-hand CP (LHCP) wave is generated. The dimensions of the dielectric plate circular polarizer are as follows (in mm): *d* = 0.9, *L*_1_ = 7.8, *L*_2_ = 28.63, *L*_3_ = 38.19, *w*_1_ = 10, *w*_2_ = 9, *w*_3_ = 2.5, *w*_4_ = 0.97, *r*_1_ = 11, *r*_2_ = 0.5, *h*_1_ = 0.762 and *w* = 0.5.

To provide a better understanding of the proposed dielectric plate circular polarizer, a parametric study of several geometrical parameters was conducted. Figure 7a shows that |S_11_| of the circular polarizer was affected by the thickness (*h*_1_) of the dielectric plate. As can be seen from Figure 7a, the selected 0.762 mm thickness of the dielectric plate corresponds to the best AR performance among the several specified dielectric plate thicknesses. As shown in Figure 7b, the simulated |S_11_| of the chosen dielectric plate thickness was better in the whole frequency band. The effect of the position of the metal via hole (*w*) on the circular polarizers AR and |S_11_| is shown in Figure 8. As shown in Figure 8, the position of the metal via hole had a great influence on the axial ratio of the circular polarizer.

Figure 9 shows the internal structure of the proposed four-level metal stepped waveguide converter, which was used to complete the interface matching between the rectangular waveguide and the circular waveguide. The dimensions of the four-level metal stepped waveguide converter were as follows (in mm): *wt*_1_ = 4.32, *wt*_2_ = 4.75, *wt*_3_ = 6.51, *wt*_4_ = 9.45, *wt*_5_ = 11, *Lt*_1_ = 10, *Lt*_2_ = 3.84, *Lt*_3_ = 3.51, *Lt*_4_ = 3.21, *Lt*_5_ = 3.65, *Lt*_6_ = 5, *rc*_1_ = 3, *rc*_2_ = 2, *rc*_3_ = 1.5 and *rc*_4_ = 1.5. As can be seen in Figure 10, the simulated bandwidth of the waveguide converter was from 18 to 32 GHz for |S_11_| ≤ −29 dB and insertion loss |S_21_| below 0.01 dB.

Figure 11 shows the influence of the second metal step width (*wt*_4_) on the performance of the converter |S_11_|. As can be seen from Figure 11, with the increase of the width of the metal step, the performance of |S_11_| gradually improved and then remained unchanged.

### 2.3. Conical Corrugated Horn Design

The conical corrugated horn antenna proposed in this design is shown in Figure 12. The proposed conical corrugated horn antenna was connected to a circular, smooth-walled input waveguide. The dominant mode of the circular waveguide was the *TE*_11_ mode, and a so-called “mode converter” was required between the circular waveguide and the conical corrugated horn. The role of the “mode converter” is to convert the *TE*_11_ mode transmitted by the circular waveguide to the *HE*_11_ mode supported by the conical corrugated horn. The dimensions of the conical corrugated horn antenna were as follows (in mm): *rh*_1_ = 14.36, *rh*_2_ = 11, *wh*_1_ = 5.38, *wh*_2_ = 3.42, *wh*_3_ = 4.62, *wh*_4_ = 6.11, *wh*_5_ = 4.68, *lh*_1_ = 2.2, *h*_6_ = 1.42, *h*_7_ = 2.92, *h*_8_ = 3.4, *h*_9_ = 2.27, *l_t_* = 11.3, *wh*_6_ = 2 and *wh*_7_ = 1.2.

The simulated results of the proposed conical corrugated horn antenna are shown in Figure 13 and Figure 14, and |S_11_| in Figure 13 is less than −15 dB in the 16.5–31.7 GHz frequency band. The maximum gain is 17.3 dBi. The simulated radiation patterns are shown in Figure 14 for both the *E*- and *H*-planes at 20, 24 and 30 GHz. The 3 dB beam bandwidth was almost the same in the *E*-plane and *H*-plane, the angular width (3 dB) was both higher than 23°, and the sidelobe levels were below −25 dB in the 17 to 30 GHz band. As the dielectric plate circular polarizer structure is symmetrical, it will not affect the performance of the horn antenna. The proposed circularly polarized horn antenna also had good rotationally symmetric radiation patterns.

To provide a better understanding of the proposed conical corrugated horn antenna, a parametric study of the length of the circular waveguide to horn antenna transition section (*l_t_*) was conducted. From Figure 15, we can see that the length *l_t_* had a relatively large impact on the |S_11_| performance of the antenna. With the increase of length (*l_t_*), |S_11_| gradually became better and then remained unchanged.

In order to verify the performance advantage of the proposed circular polarizer, another two shapes of dielectric plates in Figure 3a,b were simulated with the horn antenna. Figure 16 shows the simulation results of the circularly polarized horn antenna loaded with different circular polarizers. As can be seen in Figure 16, the |S_11_|, gain and AR of the proposed antenna were improved compared with the other two structures.

### 2.4. Design Procedure

For the convenience of the readers, a design procedure of the proposed CP horn antenna is presented as follows:Step 1: We selected the appropriate circular waveguide diameter and PCB according to the application frequency.Step 2: According to the curve equation of the improved dovetail-shaped dielectric plate, an improved dovetail-shaped dielectric circular polarizer model was established. The dielectric plate was fixed in the circular waveguide through the fixing grooves.Step 3: By adjusting the equation parameters and the thickness and length of the dielectric plate, a 90° phase difference was generated between *E_x_* and *E_y_* with equal amplitudes between them. The SIW structure was incorporated to eliminate the effect of the fixing grooves by adjusting the position of the metal via hole.Step 4: The waveguide converter was designed and optimized based on the diameter of a circular waveguide and the dimensions of a standard rectangular waveguide.Step 5: We designed a conical corrugated horn antenna based on the diameter of the circular waveguide and optimized it.Step 6: The three optimized structures were connected sequentially, including a conical corrugated horn antenna, an improved dovetail-shaped dielectric circular polarizer and a four-level metal stepped waveguide converter. The final circularly polarized horn antenna was acquired.

## 3. Results and Discussion

Based on the design above, the conical corrugated horn antenna and the dielectric plate circular polarizer were assembled and simulated. The fabricated circularly polarized conical corrugated horn antenna system after assembly is shown in Figure 17.

The simulated and measured reflection coefficients are shown in Figure 18. The measured −10 dB impedance bandwidth was 52.7% covering from 17.2 to 29.5 GHz. The simulated −10 dB impedance bandwidth was 61% covering from 17.1 to 32.8 GHz. The simulated maximum gain was 17.3 dBic at 31 GHz. 

The simulated LHCP radiation pattern and the cross-polarization (RHCP) for two orthogonal planes of the circularly polarized conical corrugated horn antenna at 20, 24 and 30 GHz are presented in Figure 19. As can be observed from Figure 19, the circular polarized horn antenna had stable directional patterns over the wide operating bandwidth. The sidelobe levels in the *xoz* plane and *yoz* plane were less than −26 dB. The cross-polarization levels in in the *xoz* plane and *yoz* plane were less than −23 dB.

## 4. Conclusions

We presented and verified a broadband circularly polarized conical corrugated horn antenna incorporating an improved dovetail-shaped dielectric plate circular polarizer. It consisted of three parts, and each part was designed and optimized independently. One of the most important parts was the design of the circular polarizer. By changing the shape of the dielectric plate, an improved dovetail-shaped dielectric plate was proposed, which was proven to be superior to the traditional dovetail-shaped dielectric plate. 

As the dielectric plate needs to be fixed inside the circular waveguide, it is necessary to fix the dielectric plate by digging fixed grooves in the inner wall of the waveguide. The performance of the circular polarizer was affected by the fixed grooves on the circular waveguide wall. In the proposed circular polarizer, a SIW structure was designed as a wall to eliminate the influence of fixed slots on the circular polarizer. Previous designs eliminated the influence of the fixed grooves on the circular polarizer by digging the compensation grooves in the plane orthogonal to the fixed grooves. 

Finally, the simulated |S_11_| of the proposed dielectric plate circular polarizer was less than −20 dB in the frequency band from 17.57 to 33.25 GHz. Then, a conical corrugated horn antenna with five corrugations and a four-level metal stepped waveguide convertor were designed and optimized. The simulated −10 dB impedance and 3 dB axial ratio (AR) bandwidths of the horn antenna integrated with the polarizer were 61% (17.1–32.8 GHz) and 60.9% (17.76–33.32 GHz), respectively. The simulated peak gain was 17.34 dBic. The measured −10 dB impedance was 52.7% (17.2–27.5 GHz).

Table 1 shows the performance comparison of the proposed horn antenna and other CP horn antenna. The proposed CP horn antenna had a better axial ratio and impedance bandwidths when compared with those seen in [5,6,7]. In addition, the proposed CP horn antenna structure is simpler, and the processing is easier.

## Figures and Tables

**Figure 1 micromachines-13-02138-f001:**
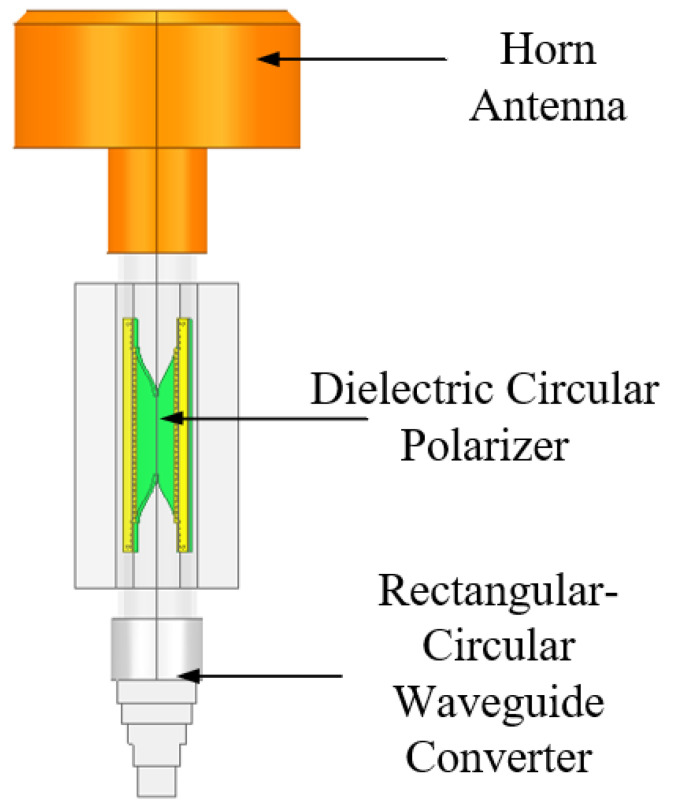
The overall structure of the proposed polarized horn antenna.

**Figure 2 micromachines-13-02138-f002:**
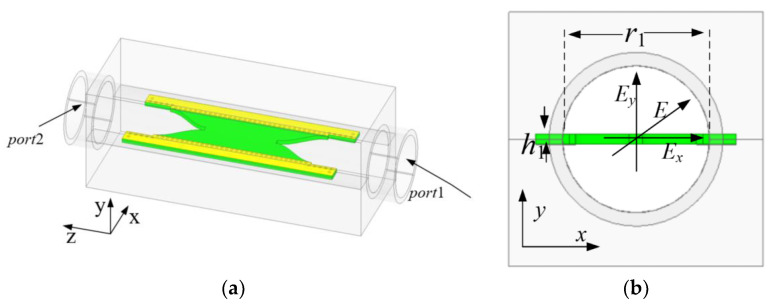
Dielectric plate circular polarizer: (**a**) Overall structure diagram. (**b**) Top view. (**c**) Cross-sectional view.

**Figure 3 micromachines-13-02138-f003:**
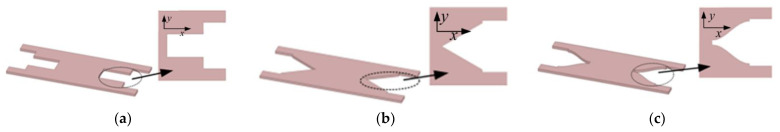
The shape of the inserted dielectric plate: (**a**) Rectangle. (**b**) Traditional dovetail shape. (**c**) Improved dovetail shape.

**Figure 4 micromachines-13-02138-f004:**
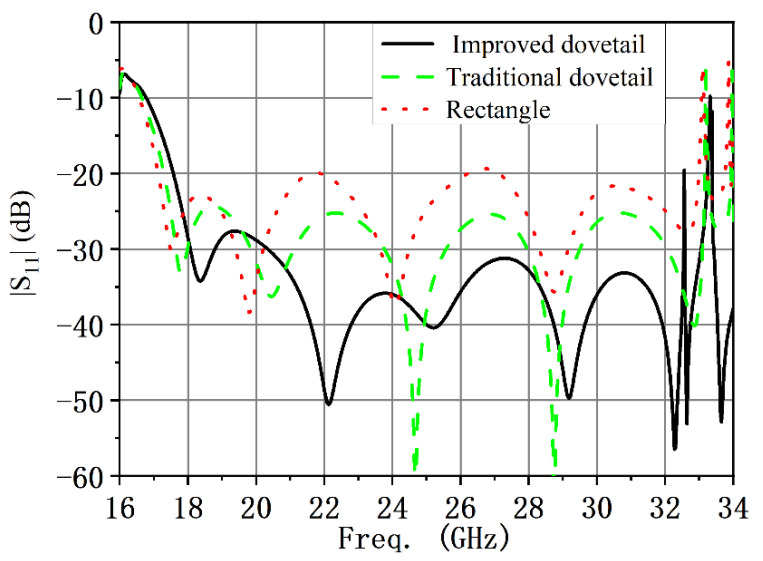
Simulated |S_11_| of three different shapes of dielectric plates.

**Figure 5 micromachines-13-02138-f005:**
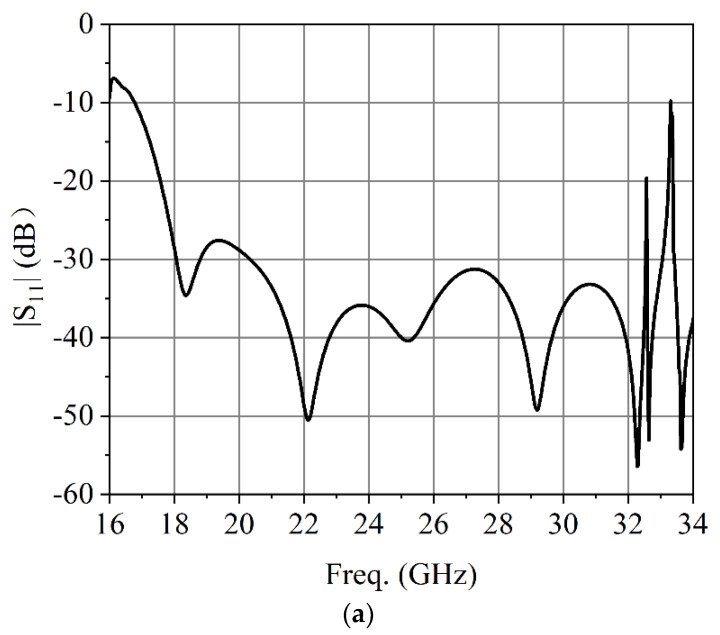
Simulated results of the dielectric plate circular polarizer: (**a**) Reflection coefficient. (**b**) AR. (**c**) *E*-field distribution in the circular polarizer at 25 GHz.

**Figure 6 micromachines-13-02138-f006:**
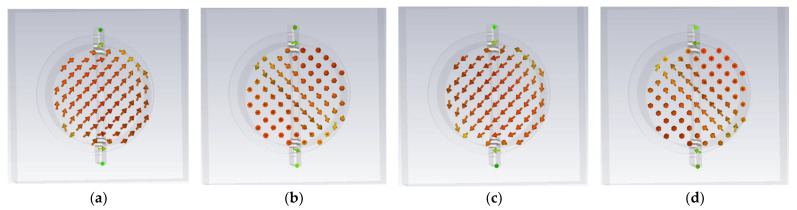
Surface current distributions on port 2 at different phases of time: (**a**) 0°. (**b**) 90°. (**c**) 180°. (**d**) 270°.

**Figure 7 micromachines-13-02138-f007:**
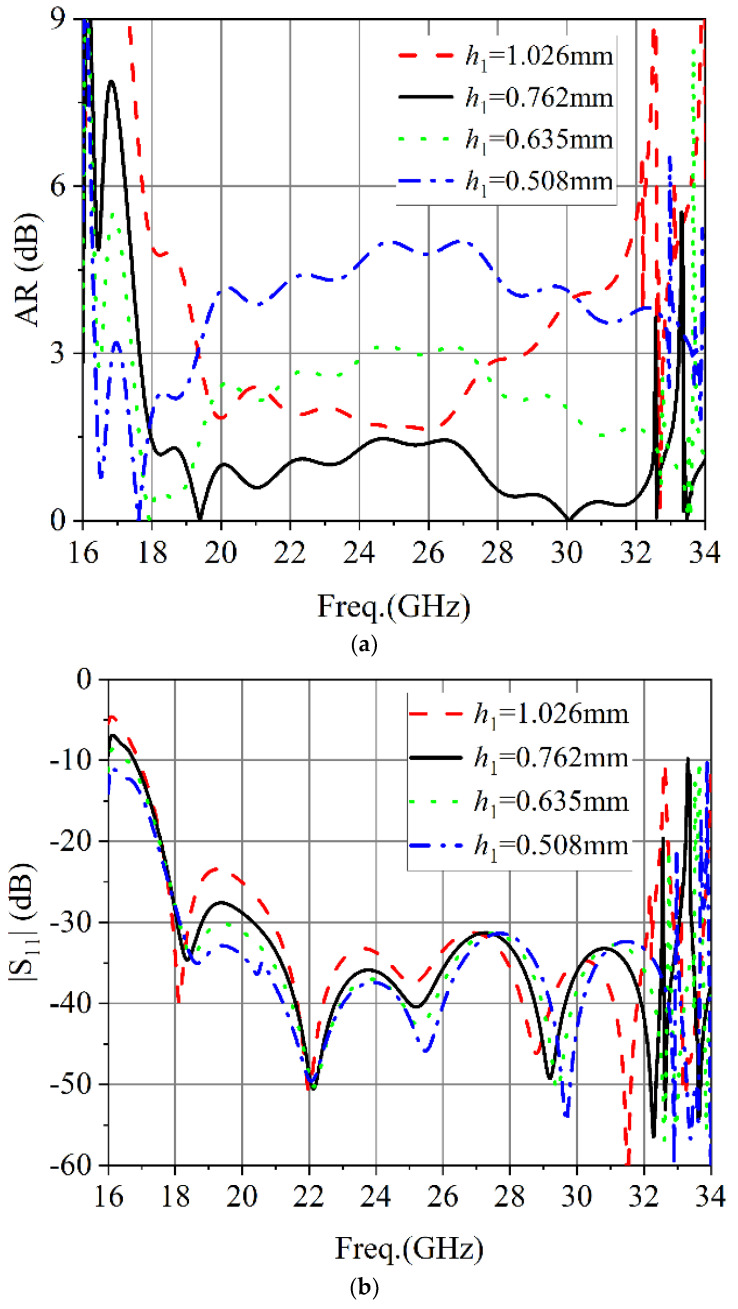
Simulated results of different dielectric plate thickness: (**a**) AR. (**b**) |S_11_|.

**Figure 8 micromachines-13-02138-f008:**
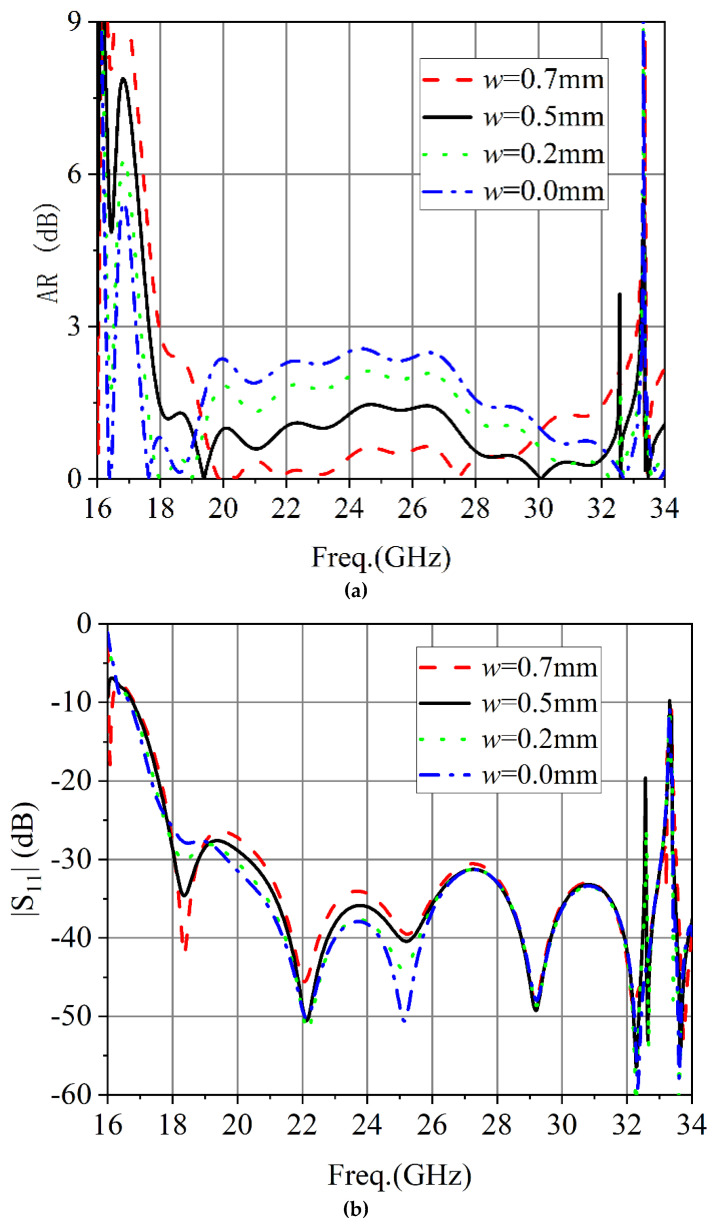
Simulated results of different hole position of the metal via hole: (**a**) AR. (**b**) |S_11_|.

**Figure 9 micromachines-13-02138-f009:**
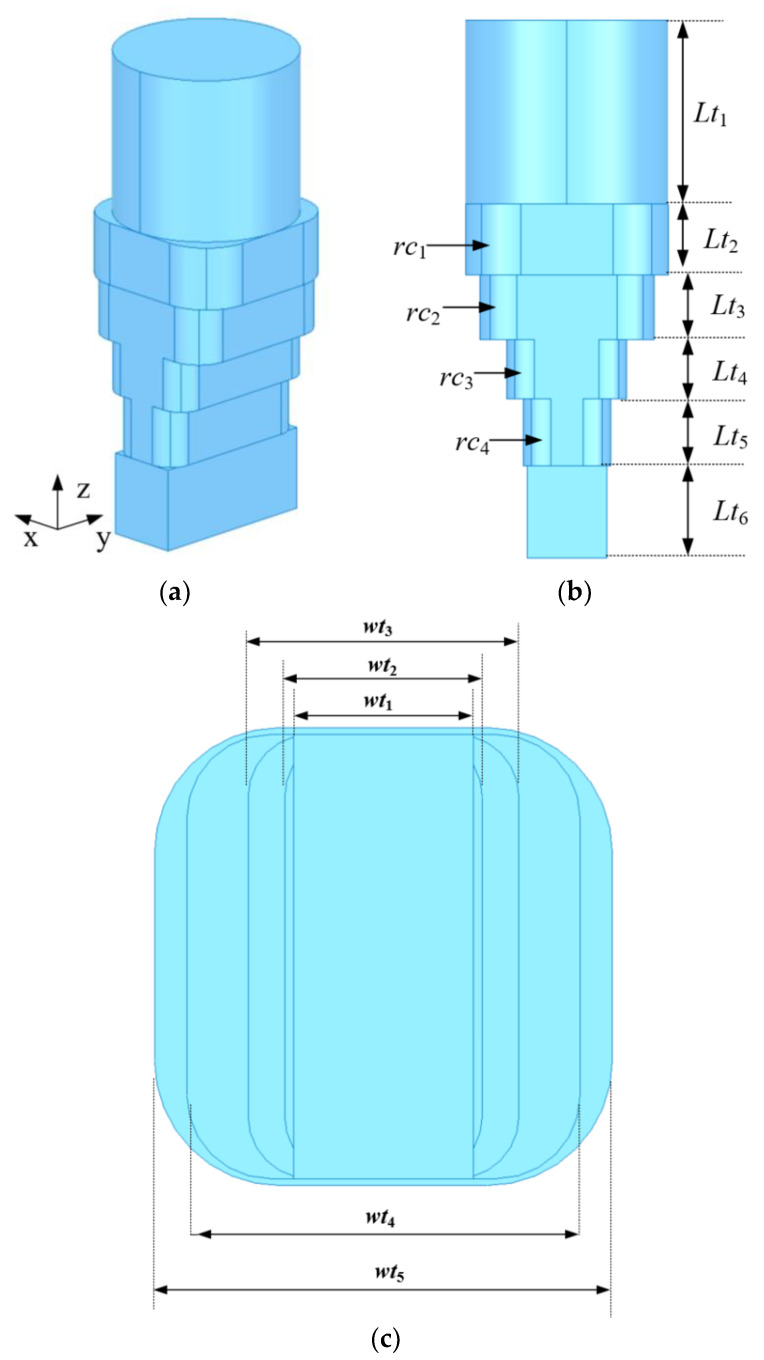
Internal structure diagram of the waveguide converter: (**a**) Isometric view. (**b**) Front view. (**c**) Bottom view.

**Figure 10 micromachines-13-02138-f010:**
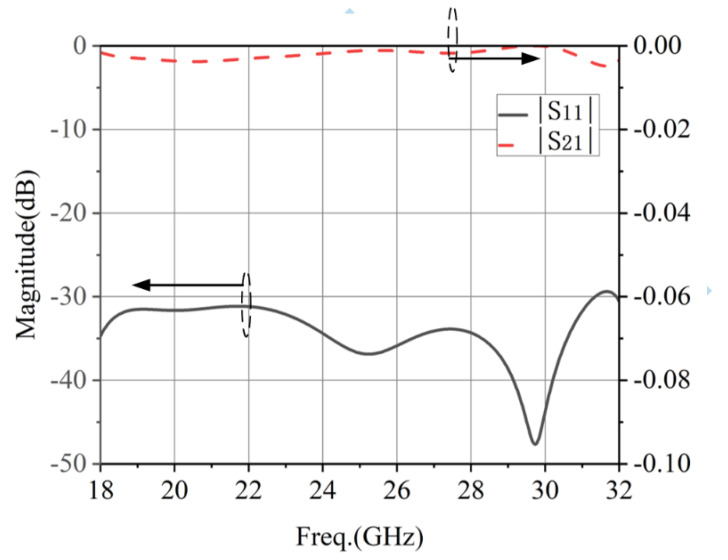
The simulated results of the four-level metal stepped waveguide converter.

**Figure 11 micromachines-13-02138-f011:**
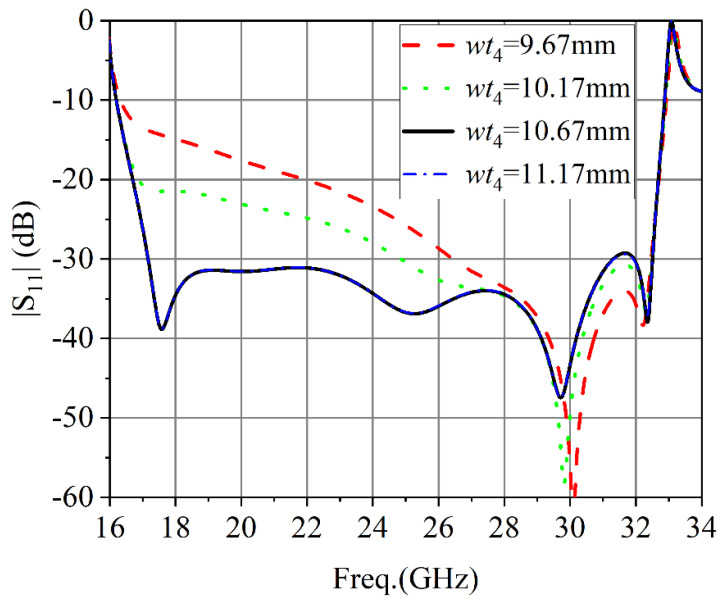
The simulated results of the second stage metal step width.

**Figure 12 micromachines-13-02138-f012:**
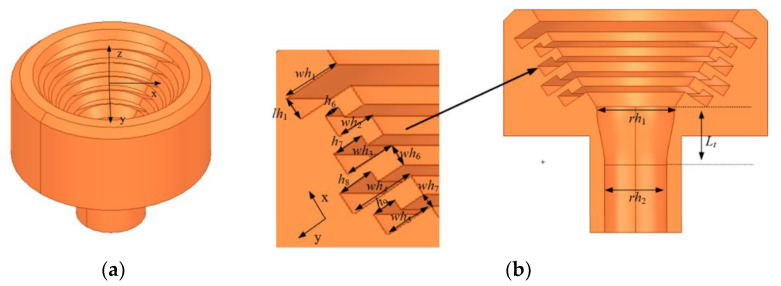
Simulated model diagram of the conical corrugated horn: (**a**) 3D view. (**b**) Cross-sectional view.

**Figure 13 micromachines-13-02138-f013:**
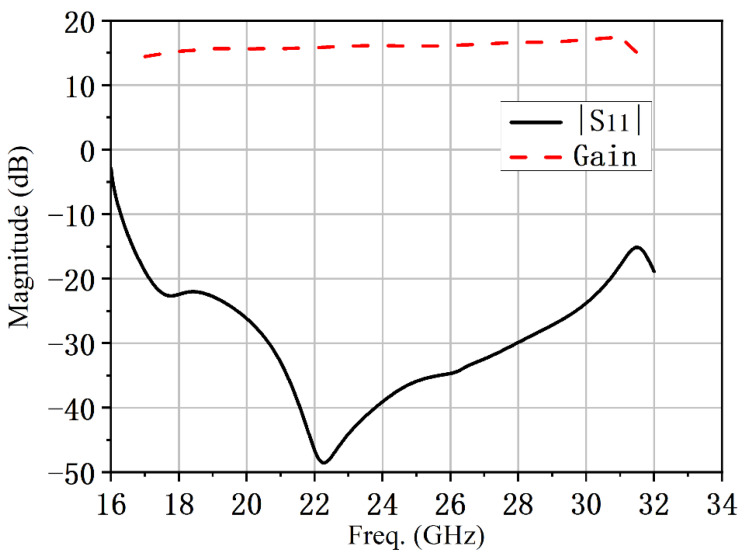
Simulated results of the proposed conical corrugated horn antenna.

**Figure 14 micromachines-13-02138-f014:**
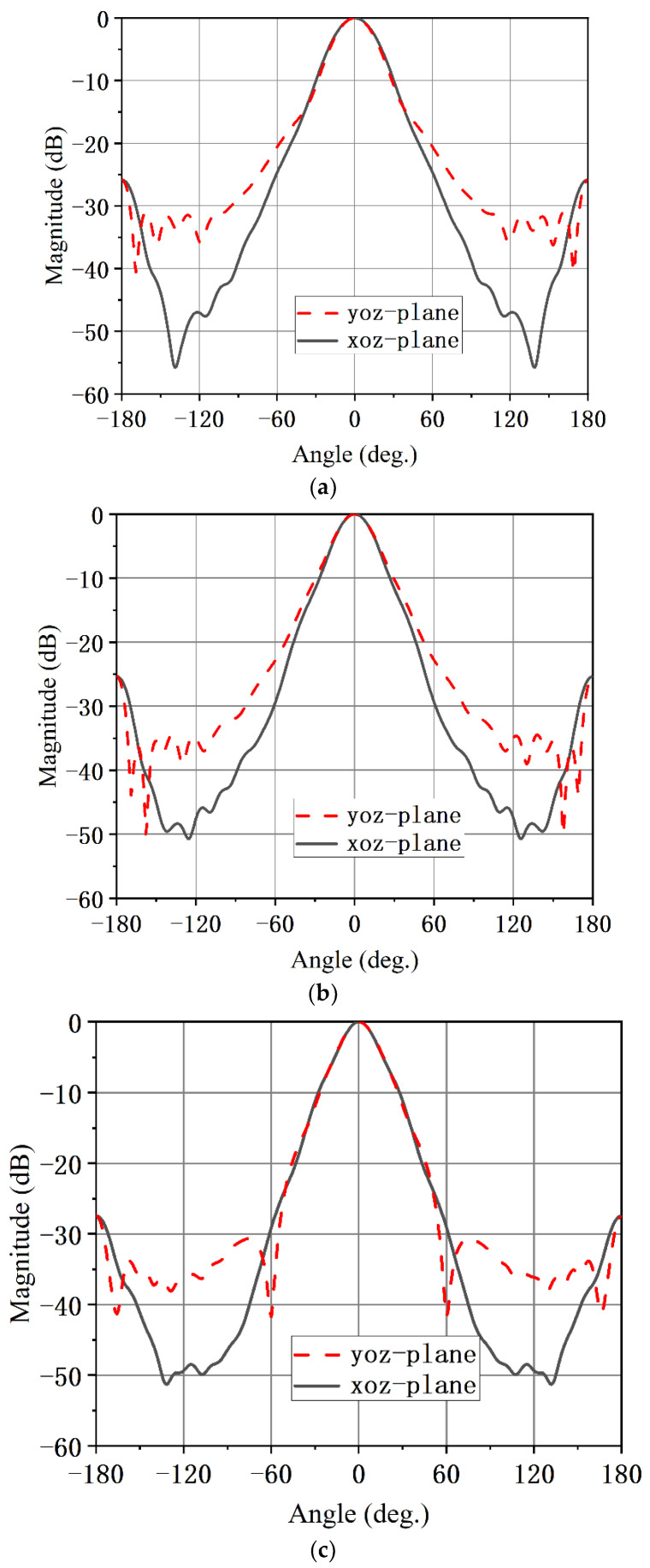
Simulated radiation patterns of the proposed conical corrugated horn antenna element at (**a**) 20 GHz, (**b**) 24 GHz and (**c**) 30 GHz.

**Figure 15 micromachines-13-02138-f015:**
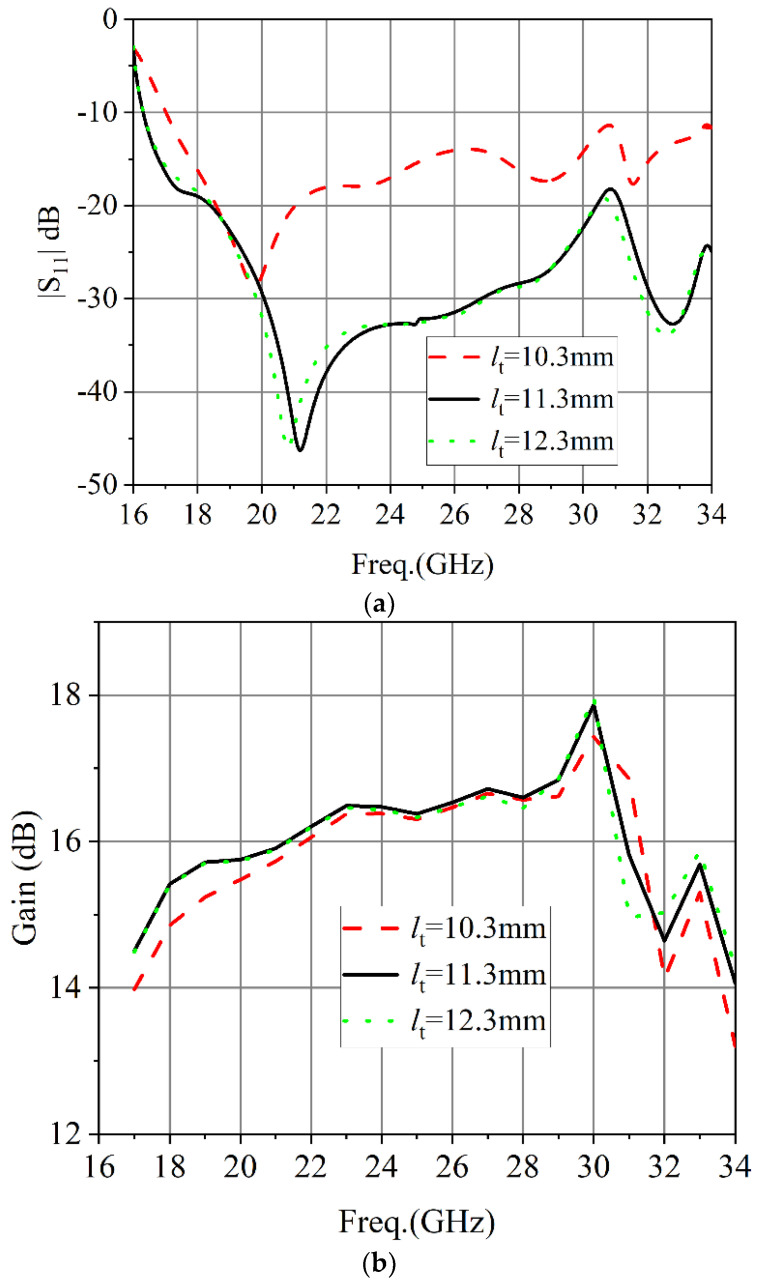
The simulated results of the length of the circular waveguide to horn antenna transition section: (**a**) |S_11_|. (**b**) Gain.

**Figure 16 micromachines-13-02138-f016:**
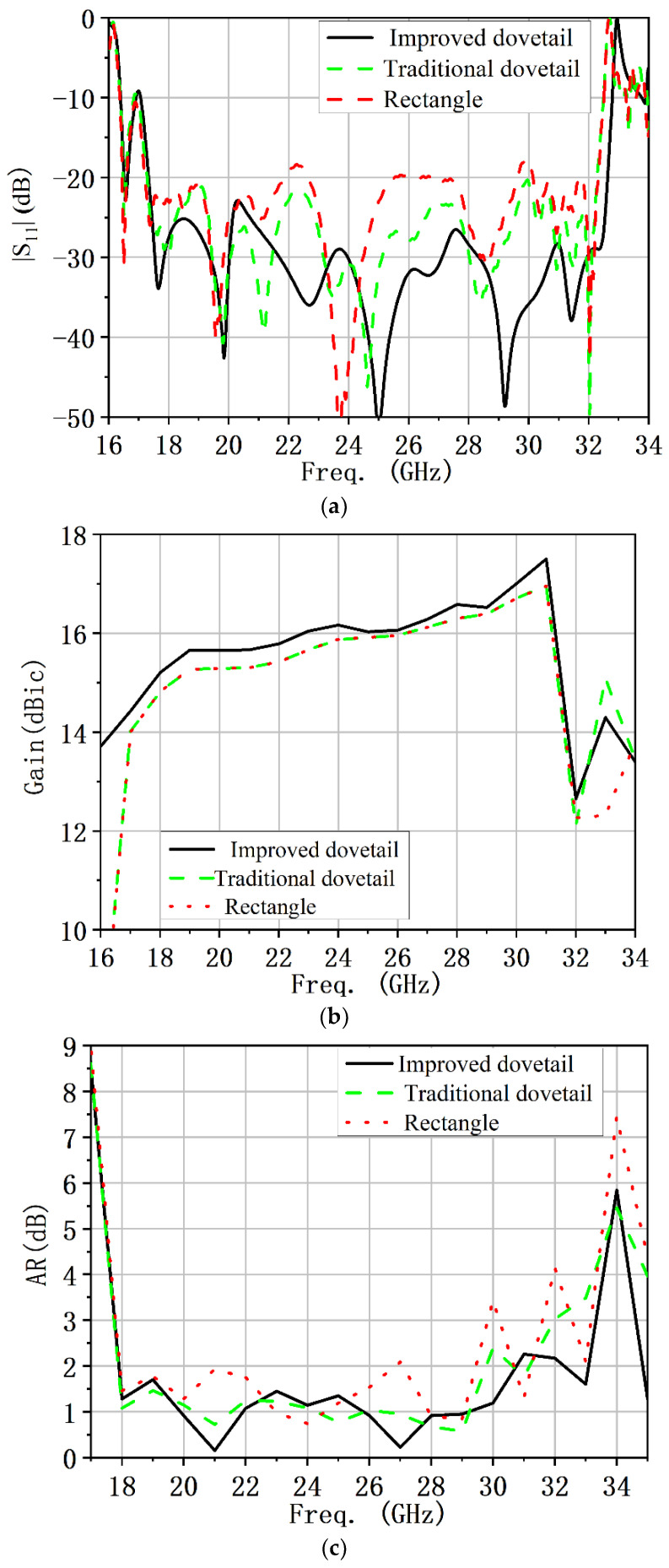
Comparison graph of simulation results of a circularly polarized horn antenna loaded with different circular polarizers: (**a**) |S_11_|. (**b**) Gain. (**c**) AR.

**Figure 17 micromachines-13-02138-f017:**
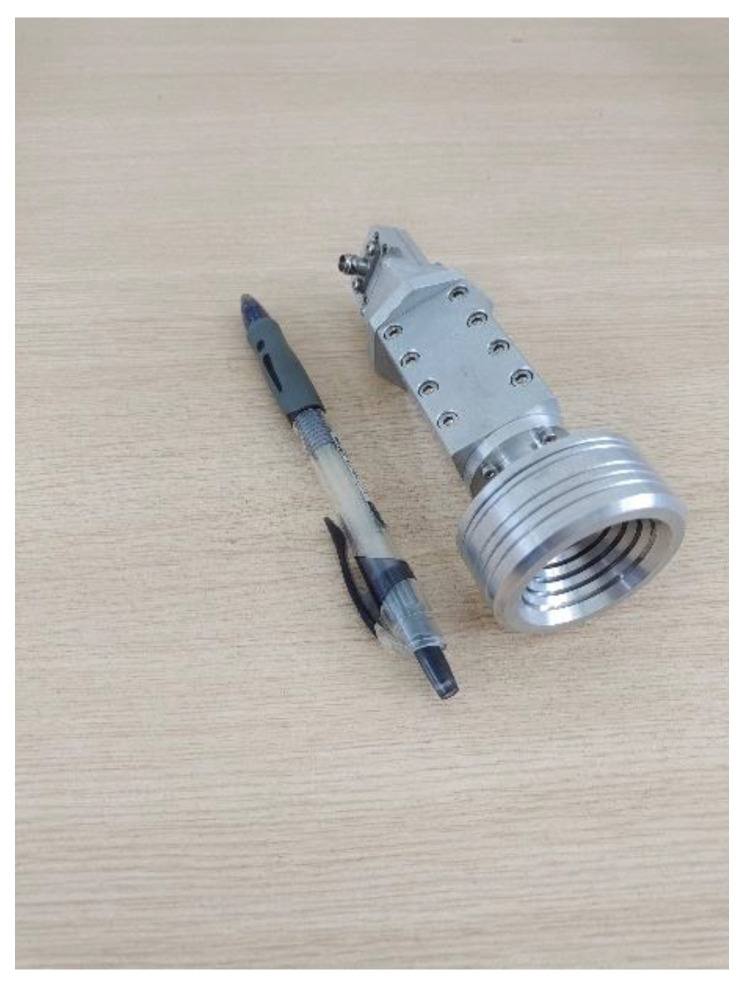
Photographs of the fabricated CP conical corrugated horn antenna.

**Figure 18 micromachines-13-02138-f018:**
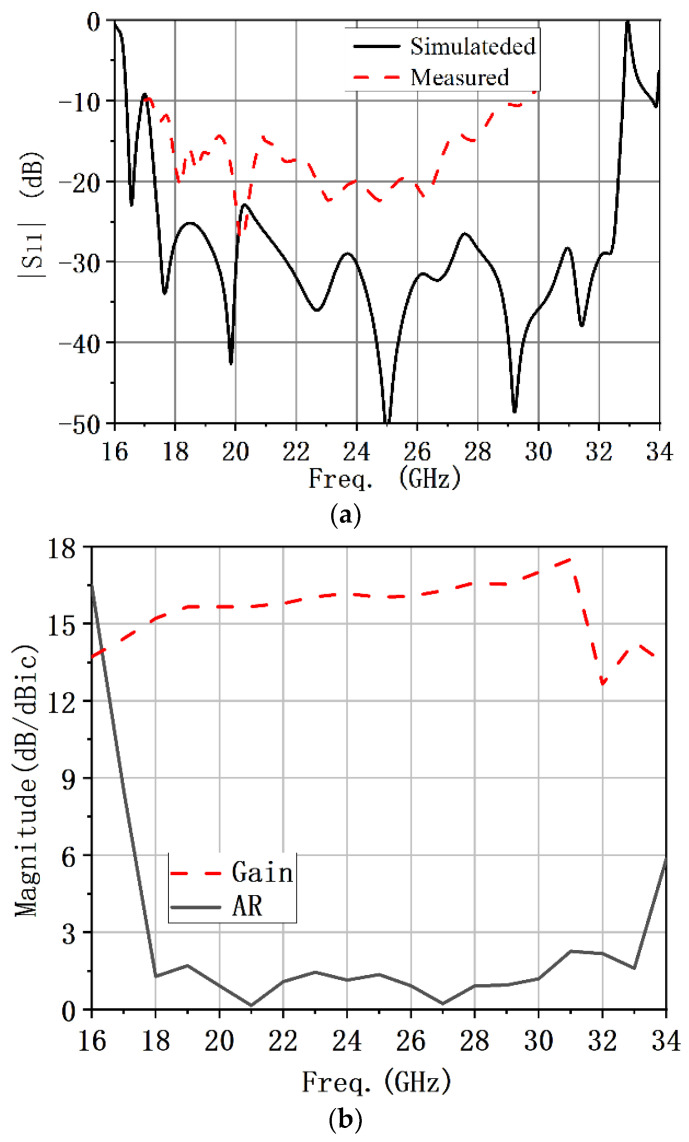
Simulated and measured results of the circularly polarized conical corrugated horn antenna: (**a**) Reflection coefficient. (**b**) Simulated gain and AR.

**Figure 19 micromachines-13-02138-f019:**
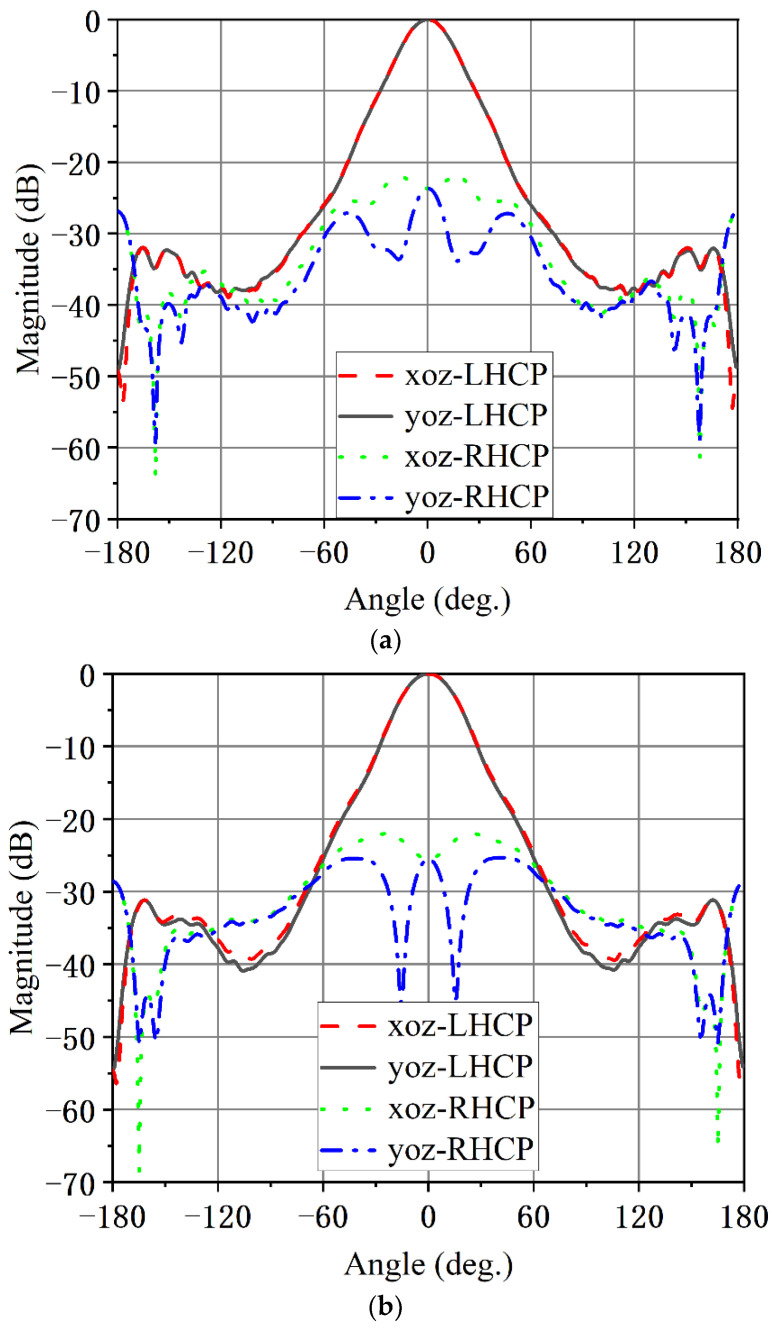
Simulated radiation patterns of the circular polarized horn antenna at (**a**) 20 GHz, (**b**) 24 GHz and (**c**) 30 GHz.

**Table 1 micromachines-13-02138-t001:** Comparisons of the antenna performance with previously proposed antennas.

Reference	Antenna Type	Impedance Bandwidth	AR Bandwidth	Peak Gain
This work	Improved dovetail-shaped dielectric plate circular polarizer and horn antenna	52.7% (17.2–27.5 GHz)	60.9% (17.76–33.32 GHz) *	17.34 dBic *
[5]	Metasurface polarizer and horn antenna	7.4% (28.5–30.7 GHz)	7.4% (28.5–30.7 GHz)	N.A. */14.2 dBic
[6]	Tapered elliptical waveguide and horn antenna	41.8% (170–260 GHz)	41.8% (170–260 GHz)	31.9 dBic */31.3 dBic
[7]	Hexagonal waveguide and horn antenna	43.4% (90–140 GHz)	37% (96–140 GHz)	N.A. */18.7 dBic

*: Simulated result. N.A.: not available.

## Data Availability

Not applicable.

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
