# Peer review of "Broadband Circularly Polarized Conical Corrugated Horn Antenna Using a Dielectric Circular Polarizer"

_micromachines, 2022, doi:10.3390/mi13122138_

Round 1

Reviewer 1 Report (Previous Reviewer 2)

The authors have made suggested corrections/improvements in the revised manuscript. I would suggest the authors make sure to check a few typos that they may find here and there in the revised manuscript.  

Author Response

Reviewer 2 Report (New Reviewer)

- The paper needs improvement, especially in content writing. The literature review needs to be added mainly for circular polarizers with dielectric plate design. I can see that the references paper is very limited and quite old. Some of the figures need to be revised because they are too small and not precise. Figure labelling is also wrong in Figure 5. Why did the authors choose this range of frequencies? The design method needs to be explained in more detail. The complete dimensions for each design need to be given in the paper.  For the measurement results, there are differences from the simulation results. Why?. Please also add a comparison and discussion between measurement and simulation for gain and AR results .

Author Response

Reviewer 3 Report (New Reviewer)

 In this manuscript Broadband Circularly Polarized Conical Corrugated Horn   Antenna Using a Dielectric Circular Polarizer is designed, simulated, fabricated and measured. The measured results validate the simulations. This is nice design with good organization. The paper can be accepted after minor modifications listed as follows:

1-Provide device under test photo.

2- The proposed device consists of three parts and in the manuscript mentioned that each part is designed and optimized independently. Provide explanations about applied optimization methods.

3- Provide sufficient explanations about comparison between the result of the traditional dovetail and proposed dovetail.

4- In the last sentence of the conclusions mentioned that the proposed device is candidate for the future 5G applications. Provide briefly explanations about 5G and three operation bands of its spectrum (low-band, mid-band and high-band). Maybe below paper is helpful: Filtering Power Divider Design Using Resonant LC Branches for 5G Low-Band Applications. Sustainability. 2022 Sep 27;14(19):12291.

5- Comparison table should be provided.

6- Radiation patterns figures with related explanations should be added.

7- Provide design procedure for these three parts of design.

Author Response

This manuscript is a resubmission of an earlier submission. The following is a list of the peer review reports and author responses from that submission.

Round 1

Reviewer 1 Report

In the presented work I did not find any improvement related to state of the art. The authors collected other wells know techniques to design a circularly polarized conical corrugated horn antenna. The only improvement is the shape of the dielectric in order to increase the matching. This is a very low result to be published. 

Reviewer 2 Report

This paper reports a broadband left-handed circularly polarized (LHCP) corrugated horn antenna using a dielectric circular polarizer. Two different designs are proposed, one where circularly polarized waves are generated by inserting a dovetail-shaped dielectric plate into the circular waveguide. In the second design, a conical corrugated horn antenna with five corrugations is illustrated. I have the following comments/concerns regarding the submitted manuscript that could potentially improve the overall quality and readability of the submitted manuscript.

1.     The authors need to clearly highlight their main contribution and novelty of their proposed experimental design over the already reported in the literature.

2.     The introduction section is too short. I suggest that the authors provide a detailed literature review on the recent developments in the circularly polarized horn antennas to justify their improved design over the existing reported work.

3.     I don’t see much difference in the design shown in Fig. 1(b), and Fig. 1(c). The authors should elaborate more on the improvement shown in Fig. 1(c).

4.     The discussion regarding Figure 3 is very superficial. The details of the overall structure should be described in detail.   

5.     In Fig. 7. There is no discussion about the S21 parameter. What useful information is provided by this parameter?

6.     It was expected that the authors would provide a detailed discussion on the simulation results provided in Fig. 10. This figure predicts some useful information that is helpful while performing the experimental validation.

7.     On page 1, line 21, “The object of this article is a circularly polarized waveguide horn antenna” the word object should be replaced by objective.

8.     Page 3 line 97, “The detailed dimensions of the four-level metal stepped waveguide conversion are follows” should be revised. conversion, as follows, is the correct way to write.

9.     The authors are expected to discuss Fig. 12 and Fig. 13 in much more detail to highlight the improvements in their design over the prior research.

10. The references count is on the low side. There are tons of papers written in this field recently, and the authors are expected to cite them.

11.  There are many typos in the manuscript that need to be fixed.
